# LEARNING RELEVANT FEATURES FOR STATISTICAL IN-FERENCE

## ABSTRACT

Given two views of data, we consider the problem of finding the features of one view which can be most faithfully inferred from the other. We find that these are also the most correlated variables in the sense of deep canonical correlation analysis (DCCA). Moreover, we show that these variables can be used to construct a non-parametric representation of the implied joint probability distribution. This representation can be used to compute the expectations of functions over one view of data conditioned on the other, such as Bayesian estimators and their standard deviations. We test the approach using inference on occluded MNIST images, and show that our representation contains multiple modes. Surprisingly, when applied to supervised learning (one dataset consists of labels), this approach automatically provides regularization and faster convergence compared to the cross-entropy objective. We also explore using this approach to discover salient independent variables of a single dataset.

## 1 INTRODUCTION

Given samples $(x_1, y_1), \ldots, (x_n, y_n)$ from an unknown joint probability distribution $p(x, y)$, we want to construct a useful representation of the conditional probabilities $p(x|y)$ and $p(y|x)$, so that we that we can infer one view from the other on new data.

For instance, $x$ and $y$ could be past and future histories of dynamical data, visual and auditory inputs, actions and their effects, etc.

Following the approach introduced in Bény & Osborne (2013; 2015b) in the context of quantum information theory, we look at the problem as follows: the conditional distributions $p(y|x)$ can be thought of as representing a noisy communication channel (stochastic map). This channel is a linear map between spaces of typically ludicrously large dimensions (the spaces of all probability distributions over $x$ or $y$). We want a pair of small subspaces which best represent the channel. Specifically, we look for those vectors representing probability distributions over $x$ which *lose least distinguishability* under the channel, where the distinguishability is measured by the $\chi^2$ divergence.

We show in Section 3 that this is equivalent to performing a certain singular value decomposition of the channel (seen as an operator in Hilbert space) and keep only the components with the largest singular values. Moreover, the full singular value decomposition is equivalent to the decomposition in terms of canonical variables introduced in Lancaster (1958), namely,

$$p(y|x) = p(y) \sum_{i=1}^{D} \eta_i u_i(x) v_i(y),$$ (1)

where $u_i$ and $u_j$ are real non-linear functions such that $\mathbb{E}(u_i u_j) = \delta_{ij}$, $\mathbb{E}(v_i v_j) = \delta_{ij}$, and $0 < \eta_D \leq \cdots \leq \eta_1 \leq \eta_0 = 1$ are the singular values.

The practical advantage of this representation for inference (or prediction) is that it reduces the evaluation of conditional expectations to that of empirical averages over the (unconditional) marginal $p(y)$.

As observed in Michaeli et al. (2016), the span of the first $k$ canonical variables $u_i$, $v_j$ is what is learned by the deep canonical correlation analysis (DCCA) (Andrew et al., 2013). Indeed, these

variables are those which maximize the correlations $\mathbb{E}(u_i v_i)$ subject to the same constraints as above. (This reduces to CCA (Hotelling, 1936) when $u_i$, $v_j$ are linear maps).

In this work, besides establishing this new information-theoretical interpretation of canonical variables and DCCA, we experiment with using this representation for performing inference on new data. Moreover, we propose a strategy for extracting disentangled variables from the canonical variables, inspired by analytical solutions.

## 2 RELATED WORK

This general problem (of building an effective representation of the conditional probability distributions implied by joint samples) covers many existing approaches in different contexts. For instance, if the variables $y$ has few possible states, then it reduces to a classification problem, usually solved by minimizing the crossentropy between a predicted distribution and the one-hot encoding of the classes.

When $y$ has a large number of states, or is fundamentally continuous, existing approaches usually do not model the whole conditional distribution, but either provide the average (regression), or approximately sample from it.

The main class of methods which allows for sampling from the conditional distributions are variational: a deterministic neural networks produces the parameters of analytical classes of probabilities. This includes variational autoencoders Kingma & Welling (2013) (e.g., Iten et al. (2018); Wang et al. (2016)), and approaches based on the minimum description length principle such as Gregor et al. (2013).

Alternatively, it may also be possible to use adversarial training Goodfellow et al. (2014), by using a conditional Mirza & Osindero (2014) version of an energy-based GAN Zhao et al. (2016).

By contrast, our approach doesn't require the training of a generative model. Instead, conditional expectations are constructed as linear combinations of unconditional empirical averages over the training data.

Previous approaches to equipping CCA or DCCA with an information-theoretic interpretation have explored different directions. For instance, in Wang et al. (2016), the authors generalize a probabilistic interpretation for CCA in terms of gaussian distributions, which leads to a variational approach. In Painsky et al. (2018), additional constraints on the mutual information between the data and the learned variables are added to the optimizations.

A previous attempt at designing a numerical solution for our singular value problem can be found in Bény (2018b). In that work, the relevant variables were represented through a PCA kernel produced by Monte Carlo sampling, but wasn't practical.

## 3 THEORY

We formalize the problem by assuming that our data was sampled from an unknown joint distribution $p(x, y)$ over two random variables $X$ and $Y$.

Let $V_X$ and $V_Y$ denote the linear spaces spanned by all probability distributions over $X$ and $Y$ respectively. Here we assume that $X$ and $Y$ take finitely many values for simplicity, but this formalism can be straightforwardly extended to infinite-dimensional vector spaces.

We will need inner products on $V_X$ and $V_Y$, to make them into real Hilbert spaces. We use the Fisher information metrics evaluated at the points $p(x)$ and $p(y)$ respectively (marginals of $p(x, y)$), that is,

$$\langle \mu, \mu' \rangle_X := \sum_x \frac{\mu(x)\mu'(x)}{p(x)} \quad \text{and} \quad \langle \nu, \nu' \rangle_Y := \sum_y \frac{\nu(y)\nu'(y)}{p(y)} \tag{2}$$

for any vectors $\mu, \mu' \in V_X$ and $\nu, \nu' \in V_Y$.

Below we also call the marginals $p_X$ and $p_Y$ respectively when omitting their arguments ($p_X(x) \equiv p(x)$ and $p_Y(y) \equiv p(y)$).

These inner products allow us to define the $\chi^2$ divergence:

$$\chi^2(q, p_X) = \langle q - p_X, q - p_X \rangle_X, \tag{3}$$

which a measures of statistical distinguishability between $q$ and $p_X$. Specifically, it quantifies how easy it is to reject the null hypothesis that the state is $p_X$ when it is actually $q$, based on the empirical distribution obtained from independent samples. It is also the lowest order approximation of the Kullback-Leibler divergence.

The joint distribution $p(x, y)$ yields conditional distributions $p(y|x)$ and $p(x|y)$. These can be understood as the components (or kernels) of stochastic maps $\mathcal{N} : V_X \to V_Y$ and $\mathcal{N}^* : V_X \to V_Y$ respectively. Explicitely, if $\mu \in V_X$ and $\nu \in V_Y$, then the images $\mathcal{N}(\mu) \in V_Y$ and $\mathcal{N}^*(\nu) \in V_X$ are defined by

$$\mathcal{N}(\mu)(y) = \sum_x p(y|x)\mu(x) \quad \text{and} \quad \mathcal{N}^*(\nu)(x) = \sum_y p(x|y)\nu(y). \tag{4}$$

These stochastic maps $\mathcal{N}$ and $\mathcal{N}^*$ perform inference of one variable given some (possibly imperfect) knowledge about the other, with priors given by the marginals $p(x)$ or $p(y)$ of $p(x, y)$ depending on the direction of the inference. Importantly, $\mathcal{N}^*$ is the *transpose* of $\mathcal{N}$ in terms of the inner products defined above:

$$\langle \nu, \mathcal{N}(\mu) \rangle_Y = \langle \mathcal{N}^*(\nu), \mu \rangle_X. \tag{5}$$

We now have the tools to address the problem mentioned in the introduction. The distinguishability between $q \in V_X$ and $p_X$ after the action of the channel $\mathcal{N}$ is $\chi^2(\mathcal{N}(q), p_Y)$ since $p_Y = \mathcal{N}(p_X)$. Hence we want to find the distributions $p$ which maximize the *relevance* Bény & Osborne (2013).

$$\eta(q) = \frac{\chi^2(\mathcal{N}(q), p_Y)}{\chi^2(q, p_X)} = \frac{\langle \mathcal{N}(\mu), \mathcal{N}(\mu) \rangle_Y}{\langle \mu, \mu \rangle_X}, \tag{6}$$

where $\mu = q - p_X$. The inner-product formulation makes it clear that this amounts to finding the eigenvector with largest eigenvalue for the symmetric map $\mathcal{N}^*\mathcal{N}$, which is also the singular vector with largest singular value for $\mathcal{N}$. On can then go on to find the eigenvector with next largest eigenvalue and so on, which are automatically orthogonal.

In practice, the inner products are more tractable to compute if we express elements $\mu \in V_X$ and $\nu \in V_Y$ in terms of variables $f$ and $g$ as $\mu = p_X f$ and $\nu = p_Y g$, or

$$\mu(x) = p(x)f(x) \quad \text{and} \quad \nu(y) = p(y)g(y)$$

for all $x, y$. Indeed, this yields simply

$$\langle \mu, \mu' \rangle_X = \mathbb{E}(ff') \quad \text{and} \quad \langle \nu, \nu' \rangle_Y = \mathbb{E}(gg').$$

We are now in measure to make the connection with DCCA Andrew et al. (2013). Indeed, the aims of DCCA is to maximize the correlations $\text{corr}(f, g) = \mathbb{E}(fg)$ over function $f(x)$ and $g(y)$ such that $\mathbb{E}(f^2) = \mathbb{E}(g^2) = 1$. But, using $\mu = p_X f$ and $\nu = p_Y g$, we have

$$\mathbb{E}(fg) = \langle \nu, \mathcal{N}(\mu) \rangle_Y,$$

which is maximized by the left- and right- singular vectors $\mu$ and $\nu$ of $\mathcal{N}$ with largest singular value.

Given all the singular vectors $\mu_i = p_X f_i$ and $\nu_i = p_Y g_i$ with singular values $\eta_i$, we obtain the representation

$$\mathcal{N}(\mu) = \sum_i \eta_i \nu_i \langle \mu_i, \mu \rangle_X,$$

which, using a more standard notation and the Kronecker delta $\delta_x$, yields Eq. 1:

$$p(y|x) = \mathcal{N}(\delta_x)(y) = \frac{1}{p(x)} \sum_i \eta_i \nu_i(y)\mu_i(x) = p(y) \sum_i \eta_i v_i(y)u_i(x),$$

where $\mu_i = p_X u_i$ and $\nu_i = p_Y v_i$.

### 3.1 NON-DIAGONAL FORM AND RELEVANT VARIABLES

For the purpose of the optimization and inference, we do not need to full diagonal decomposition, but just functions $f_i = \mu_i/p_X$ and $g_j = \nu_j/p_Y$, $i,j = 1,\ldots,k_0$, which have the same span as the canonical variables $u_i$ and $v_j$ respectively for $i,j = 1,\ldots,k_0$ (assuming that $\eta_1,\ldots,\eta_{k_0}$ are the largest singular vector). Below we refer to $f_i$ and $g_j$ as the $k_0$ *most relevant variables*.

Because these functions may not be orthogonal, we need the covariance matrices

$$K_{ij} = \langle \mu_i, \mu_j \rangle_X = \mathbb{E}(f_i f_j), \quad L_{ij} = \langle \nu_i, \nu_j \rangle_X = \mathbb{E}(g_i g_j), \quad A_{ij} = \langle \nu_i, \mathcal{N}(\mu_j) \rangle_Y = \mathbb{E}(g_i f_j).$$

If $N_{ij}$ denote the components of $\mathcal{N}$ in the sense that $\mathcal{N}(p_X f_j) = \sum_i N_{ij} p_Y g_i$, then, using our inner products to isolate $N_{ij}$, we obtain $N = L^{-1}A$. Similarly, the components of $\mathcal{N}^*$ are $N_{ij}^* = K^{-1}A^\top$. This implies that the sum of the square of the singular values of $\mathcal{N}$ restricted to the spans of the vectors $p_X f_i$ and $p_Y g_j$ for all $i,j$, which is what we want to maximize, is just given by

$$\sum_{i=1}^{k_0} \eta_i^2 = \mathrm{Tr}\,(N^* N) = \mathrm{Tr}\,(K^{-1}A^\top L^{-1}A).$$

This is the DCCA objective. Below we use the objective function $C = k_0 - \mathrm{Tr}\,(N^* N)$, for the cosmetic reason that its optimal value is zero.

Moreover, the corresponding truncated representation of the conditional distribution is

$$p(y|x) = \mathcal{N}(\delta_x)(y) \simeq p(y) \sum_{i,j=1}^{k_0} (L^{-1}AK^{-1})_{ij} g_i(y) f_j(x),$$

where we used the fact that the components of $\delta_x$ are $\delta_j = \sum_i K_{ji}^{-1} f_i(x)$.

Of course, This approach can produce a faithful representation of the correlations only if $\mathcal{N}$ is actually close to being of rank $k_0$ (see Appendix B for a more precise statement). If we interpret the relevant subspace as a space of probability over latent variable, this means that our latent variables have at most $k_0$ discrete states.

However, even if the rank $k_0$ corner of $\mathcal{N}$ is a not a good approximation, this strategy allows us to nevertheless do the correct inference on certain random variables, namely those which are in the span of the canonical variables!

Indeed, the exact conditional expectation of $g_k$ is (assuming $D$ is the actual rank of $\mathcal{N}$),

$$\sum_y g_k(y) p(y|x) = \sum_{i,j=1}^{D} (L^{-1}AK^{-1})_{ij} \mathbb{E}(g_k g_i) f_j(x)$$

$$= \sum_{j=1}^{D} (AK^{-1})_{kj} f_j(x) = \sum_{j=1}^{k_0} (AK^{-1})_{kj} f_j(x),$$

where the last truncation is exact if $k \leq k_0$ due to the assumption that the basis $f_i$ and $g_j$ have the same span as the $k_0$ largest right and left singular vectors of $\mathcal{N}$ respectively.

For instance, if $p(x,y)$ is Gaussian, the canonical variables can be computed analytically, as in Lancaster (1958) or Bény (2018a) in the multivariate case. Solutions for other distributions were also computed in Eagleson (1964).

Notably, for any two dimensional Gaussian, the space of $k$ most relevant variables is simply spanned by the moments $f_n(x) = x^n$ and $g_n(y) = y^n$ for $n = 0,\ldots,k-1$. Hence, in this case the first $k$ moments can be inferred exactly using only the $k+1$ most relevant variables (See Appendix C).

## 4 ALGORITHM

Let us explicit the algorithm resulting from the above analysis.

We assume that we are given independent samples $(x_1, y_1), (x_1, y_2), \ldots$ from the otherwise unknown joint distribution $p(x, y)$.

We first perform DCCA (Andrew et al., 2013). That is, we need two *independent* deterministic feed-forward neural networks. The first maps $x$ to a set of $k_0$ real-valued variables $f_1(x), \ldots, f_{k_0}(x)$. The second maps $y$ to a different set of $k_0$ variables $g_1(y), \ldots, g_{k_0}(y)$.

The parameters of the neural networks are to be set to minimize the objective function

$$C = k_0 - \mathrm{Tr}\,(K^{-1} A^\top L^{-1} A), \tag{7}$$

where the matrices $K, L, A$ can be approximated over a mini-batch $(x_n, y_n), n = 1, \ldots, N$ via

$$K_{ij} = \frac{1}{N} \sum_{n=1}^{N} f_i(x_n) f_j(x_n), \quad L_{ij} = \frac{1}{N} \sum_{n=1}^{N} g_i(y_n) g_j(y_n), \quad A_{ij} = \frac{1}{N} \sum_{n=1}^{N} g_i(y_n) f_j(x_n). \tag{8}$$

We found that, provided the batch size is sufficiently large compared to $k_0$ (about 10 times in our experience), this can be minimized using ADAM or direct gradient descent. However, to guarantee stability when using large $k_0$, we needed to explicit the gradient of the objective function in order to force the use of the Moore-Penrose pseudo-inverses for $K^{-1}$ and $L^{-1}$ in both the forward and backward passes, in addition to using 64 bits floats in these computations.

Once the relevant variables have been learned, we still need to use the training data in a second step. Indeed, suppose that we wish to use our model to infer the value of some function $\Theta(x)$, i.e., to compute its approximate expectation value in terms of the conditional distribution $x \mapsto p(x|y)$. Then we need to store, for each variable $j = 1, \ldots, k_0$, the quantities

$$\Theta_j = \frac{1}{N_{\mathrm{full}}} \sum_{n=1}^{N_{\mathrm{full}}} \Theta(x_n) f_j(x_n), \tag{9}$$

where the average is to be taken on the full training batch (of size $N_{\mathrm{full}}$). The same can be done exchanging $x$ with $y$ and $f_j$ with $g_j$ for the reverse inference.

For instance, if a data point $x$ is composed of real components $x^a$—such as pixel color components for an image—and we are interested in the estimator which minimize the expected $l^2$ distance to the predicted values of these components, then we need the expectation values of the components $\Theta(x) = x^a$ for all $a$, and possibly higher moments to gain more knowledge about the shape of the posterior distribution, such as the second moments $\Theta'(x) = x^2$, etc.

Inference can then be performed with new data using

$$\overline{\Theta} = \sum_x p(x|y)\,\Theta(x) \approx \sum_{i,j=1}^{k_0} (K^{-1} A^\top L^{-1})_{ji} \Theta_j g_i(y). \tag{10}$$

Moreover, the accuracy of these predictions does not depend on the rank $k_0$ if $\Theta$ is taken in the span of the relevant variables, i.e., $\Theta(x) = \sum_{i=1}^{k} c_i f_i(x)$, for which $\Theta_j = \sum_{i=1}^{k_0} c_i K_{ij}$.

The reverse inference formulas are obtained simply by the exchanges $K \leftrightarrow L$, $A \leftrightarrow A^\top$, and $g_i \leftrightarrow f_i$.

## 5 EXPERIMENTS

In all our experiments, we used the ADAM optimizer with learning rate $0.001$. We used the Flux package (Innes, 2018) for Julia, as well as Tensorflow.

As usual the data is divided into a training set and a testing set. No aspect of the testing set is used during training. The loss function refers to Eq. (7). In order to monitor overfitting, we compute a "test loss" and a "training loss". The test loss is computed from the trained variables using only the test data, and accordingly, the training loss is computed purely using the training data.

Moreover, when performing inference on test data using Eq. (10), we use the covariances $A, L, K$ and expectations $\Theta_j$ (Equ. (9)) built from the training data only.

## 5.1 Inference on occluded MNIST

In this experiment, we use the left and right halves of the MNIST digit images as correlated variables $X$ and $Y$. The goal is to obtain the expected left halves given the right halves, or vice versa.

The training set was augmented by random small rotations and displacements to make the task more ambiguous, as we want to explore the uncertainty in the prediction.

The relevant variables were represented by two convolutional neural networks of identical architecture. They are composed of four convolutional layers and one fully connected layer, an architecture that performs well for supervised learning on this dataset. For ease of implementation, these CNN have the whole image as input, but with either half zeroed (same value as black pixels). Half-width CNNs with proper padding at the cut perform similarly.

After training, we used the training dataset to also compute the expected pixel gray value as well as their covariance for each relevant variable using Eq. (9).

These were used into Eq. (10) to compute the mean pixel gray values and their covariances over the conditional probability of $X$ given $Y$ on test data. This mean is the Bayesian estimator for the $l^2$ distance between half images, i.e., it should minimize the expected distance $d_{l^2}$ over the conditional distribution, where $d_{l^2}^2(x, y) = \sum_i (x_i - y_i)^2$, where $x_i \in [0, 1]$ is the value of the $i^{th}$ pixel. (This is equivalent to minimizing the mean square error).

The results on a randomly selected subset of test digits is shown in Fig. 1. For each example, we also computed the images obtained by adding plus or minus one standard deviation along the direction of greatest variance in the space of relevant variables. This reveals the main ambiguities (such as between $8$ and $3$ or $7$ and $9$ which share a similar right half).

The graph of the singular values shows that the rank cutoff of 200 is too low to capture all of the relevant variables (the sudden drop at the end is not robust to an increase in the cutoff), but the results are reasonable nevertheless. This shows that our representation of the conditional distributions contains valuable information besides the simple mean.

## 5.2 Supervised learning

In the context of a supervised classification task, one of the dataset (the labels) is of sufficiently low dimensionality that we can use a complete basis over its probability space as our relevant variables, such as the standard one-hot encoding of labels. This serves as a good first sanity test for our approach. Surprisingly, we find that it converges faster than standard approaches, and without the need for regularization.

Let the variable $Y$ stands for the labels, with values in $\{1, \ldots, k\}$. The probability space consists of vectors with $k$ real components. The canonical basis corresponds to the one-hot encoding $g_i(j) = \delta_{ij}$ (Kronecker delta). All we need is a neural network to encode $k$ variables $f_1, \ldots, f_k$ on $X$. After learning the most relevant variables $f_i$, we apply the reverse of Eq. (10) for function $\Theta(y) = y$, and use the maximum component of expected value $\overline{y}$ to infer the labels from the data.

Let us refer to this procedure as DCCI (Deep Canonical Correlations based Inference).

We tested this approach on the MNIST and CIFAR10 datasets, and compared the results to the standard cross-entropy objective (Fig. 2).

We plotted the accuracy as function of the epoch rather than clock time which would depend on many factors. But the time per epoch is roughly the same for each approaches in the above experiments. Indeed, the training time is dominated by the forward and backward evaluations of the neural networks which are identical. (However, the time it takes to evaluate our objective can become significant for much larger number of labels $k$, since it involves the inversion of matrices of dimension $k$. This is in addition to the fact that a greater dimension would require also larger batches.)

We found that, without regularization, simply changing the objective from cross-entropy to DCCI provided a large improvement both of convergence speed and final accuracy for both models.

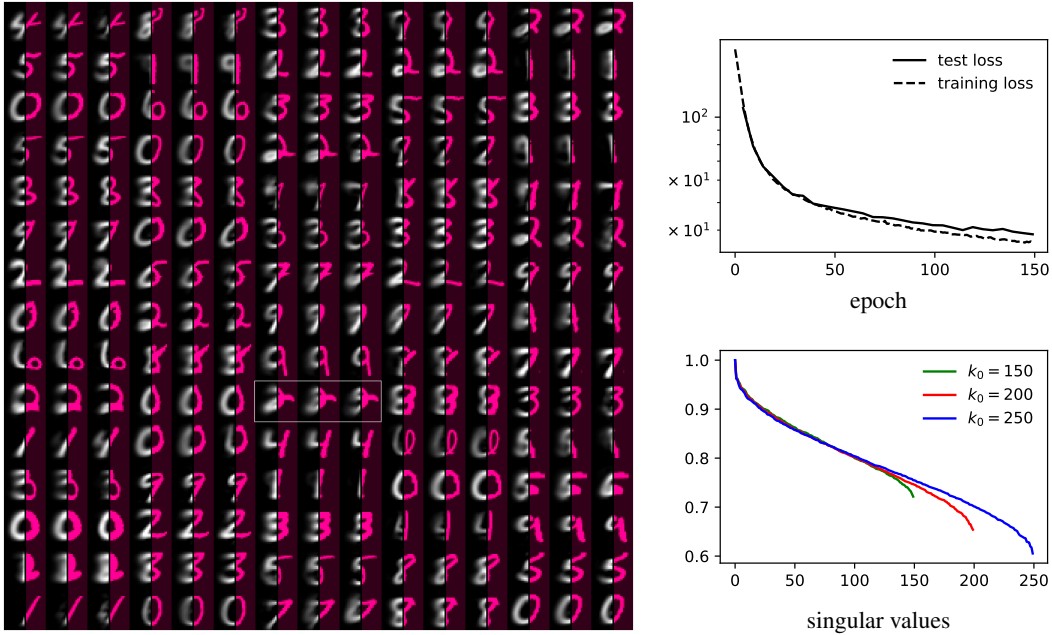

Figure 1: Left mosaic: the left halves of the MNIST digits in a random sample from the test set are inferred from the right half, with cutoff $k_0 = 200$. Three images are shown for each digit. Within each triplet, the middle image represents the mean over pixel intensity of the inferred condition distribution, while left and right images corresponds to a plus and minus one standard deviation from the mean in the direction of largest covariance (in the space of half-images). A particularly interesting example is highlighted. Top-right: loss per epoch for $k_0 = 200$. Bottom-right: the singular values for different values of the cutoff $k_0$, after 150 epochs in each case.

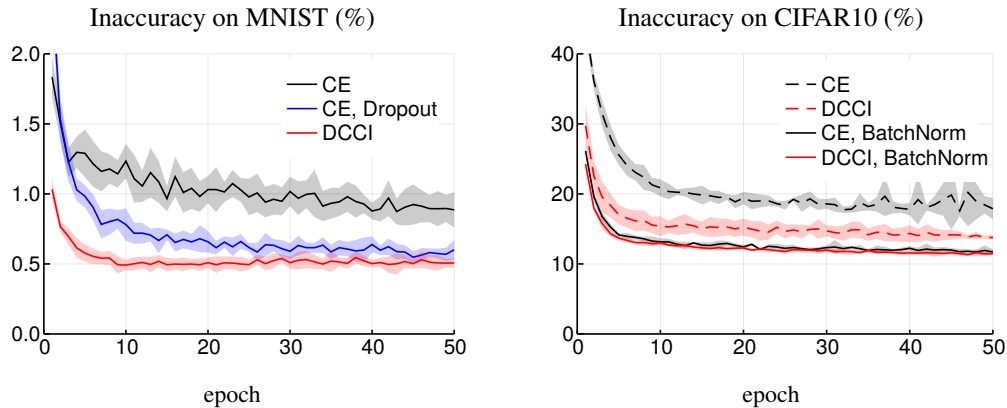

Figure 2: Loss and inaccuracy (error rate) on test sets for two classification tasks. The models were trained either using the cross-entropy (CE) or our approach (DCCI), with or without regularization layers. On the MNIST dataset, we used an "all CNN" network, and for the CIFAR10 dataset we used a short VGG variation with 10 convolutions and 3 fully connected layers. In the regularized form, post-activation Batchnorm layers were placed after each convolutional layers on the VGG network. What is shown is the mean over 10 independent runs for MNIST and 5 runs for CIFAR10. The shaded area spans the standard deviation. ADAM with default parameters was used in all cases. No data augmentation was used except for horizontal flips for CIFAR10 (resulting in epochs of 100,000 images).

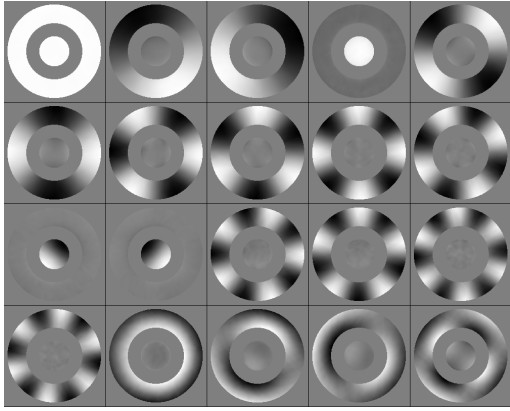 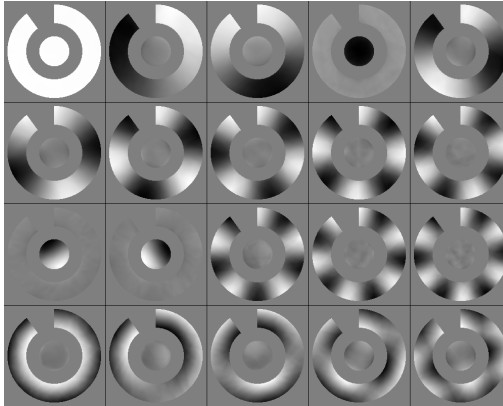

Figure 3: *Top-left*: First 20 relevant variables (on $X$) determined by DCCA for a system where $X$ consists of two coordinates uniformly sampled over a circle and a surrounding ring, and $Y$ consists of the same points but shifted by a small normally distributed vector. The variables are arranged from left-to-right and top-to-bottom in order of decreasing relevance. *Top-right*: the same variables multiplied by the marginal $p_X$. *Bottom row*: introducing a gap in the ring allows for a monotonous function of the angle to serve as second most relevant variable (instead of the sine/cosine couple). Hence the angle is automatically "disentangled" from the other variables. (Mid-gray represents the value 0).

On MNIST, DCCI alone also outperformed cross-entropy with dropout. (Dropout did not yield any improvement in conjunction with DCCI). However, adding batch-normalization layers on the CIFAR example, erased any distinction between DCCI and cross-entropy.

### 5.3 STRUCTURE OF THE RELEVANT VARIABLES

We mentioned in Section 3 that if $p(x, y)$ is a two-dimensional Gaussian distribution with zero mean, then the $n$ most relevant variables of $X$ are the first $n$ powers of $X$ itself, independently of the covariance matrix. This implies that the canonical variables are the Hermite polynomials in $X$ (which results from applying the Gram-Schmidt procedure to the basis $\{1, x, x^2, \dots\}$).

A similar property holds for multivariate Gaussians, namely, the less relevant singular values are polynomials in the more relevant ones. If this is true more generally, it should be possible to further compress and organize the latent space extracted with DCCA by finding a minimal set of generators, which ought to also be in the span of the most relevant variables.

We applied DCCA to a synthetic dataset to explore this idea, shown in Fig. 3. In this case, we actually performed a final SVD to obtain the unique uncorrelated canonical variables, and ordered them by decreasing relevance (their respective singular values).

Here, $X$ consists of two real numbers, distributed uniformly within a ring and a disk. The variable $Y$ is obtained by adding a random Gaussian shift to $X$ with a small standard deviation. The more relevant variables ought to be those which are more robust to such small random displacement. This formalizes the idea that we are interested in extracting "large-scale" variables Bény (2018b).

We would expect the relevant independent variables to be: the binary variable indicating whether the point is in the disk or the ring and the angle around the ring, followed by the radial component in the ring, and finally the Cartesian coordinates inside the disk. This is precisely what we see in Fig. 3.

Indeed—if we put aside for now the fact that the angle itself is not directly represented—besides the constant function, the two most relevant variables are the sine and cosine of the angle, followed by the binary variable separating the disk from the ring.

But these variables ought to span the space of probabilities over the relevant variables, not just the variables themselves. Hence the next six variables are sines and cosines of smaller wavelength,

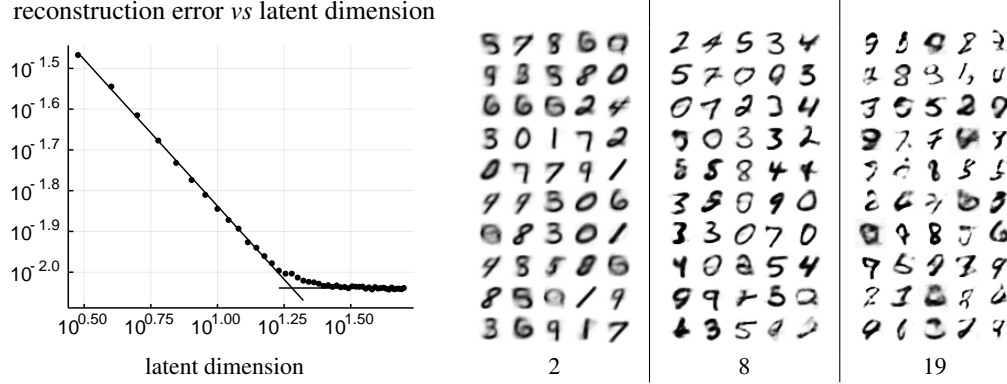

Figure 4: Left: Best mean squared error for images reconstructed from the $k$ most relevant variables, as a function of $k$ (the latent dimension). This logarithmic plot shows that improvements stop once the dimension reaches 19 (where the two lines cross). Right: images produced by the generator from latent variables sampled according to the best Gaussian fit in latent space, for feature subspaces of dimensions 2, 8 and 19.

which can encode probability distributions which are increasingly more precisely localized, down to a precision (wavelength) comparable with the diameter of the inner disk. Accordingly, the next two most relevant variables are the Cartesian coordinates inside the disk. This is followed by additional moments of the angle, down to a wavelength equal to the ring's thickness, at which point we see the radius in the ring appear.

As mentioned, we see that the angle itself is not represented, likely because it is discontinuous. However, as shown also in Fig. 3, creating a gap in the ring allows for the angle to emerge as most relevant variable. This suggests that this approach may be able to automatically learn intrinsic coordinates of the latent variable manifold.

## 5.4 INDEPENDENT VARIABLES AND GENERATIVE MODEL

If we postulate that the independent (or disentangled) relevant latent variables can be found in the linear span of the relevant variables, we can attempt to extract them by optimizing a neural network composed of two parts. Firstly, a linear layer maps the relevant variables to a small number of outputs (equal to the *latent dimension*). The purpose of this linear layer is to find the independent variables. These latent variables are then processed by an arbitrarily complex generative network to produce a possible value of the variable $X$. As objective function, we may us an appropriate measure of similarity between the output and the data element from which the variables were obtained.

We tested this idea as follows. We took $X$ to consist of the MNIST digits, and produced $Y$ by randomly permuting neighboring pixels in the image, until the mean displacement per pixel is of order 1. In addition, we added independent Gaussian noise to the pixel values. (Hence the noise map $\mathcal{N}$ simulates the coarse-graining channel introduced in Bény & Osborne (2015a)).

As in the previous experiment, we do so to implement our intuition that the more relevant variables ought to be the ones which are of larger scale, or more robust to local perturbations.

The relevant variables of the clean images were produced by the same convolutional neural network as in Section 5.2, while the variables of the coarse-grained images were extracted by a network of the same geometry, but with half the number of filters and neurons.

We extracted the 1000 most relevant out of 1200 learned variables in this way. (The least relevant variables in this system happen to be highly dependent on the total number of variables and hence cannot be trusted to be correct). As a second step, we trained a linear layer coupled to a network composed of 5 fully-connected layers of 800 hidden neurons each. We refer to the number of output neurons in the first linear layer as the *latent dimension*.

As input, this network received the variables extracted from MNIST images using the above convolutional neural net (after it was fully trained using DCCA), and was trained to minimize the mean square error between its output and the original MNIST digit.

The resulting best mean square errors are shown in Fig. 4, as function of the latent dimension. Here we see a distinct change of polynomial scaling law at dimension 19. Increasing the dimension further provides no improvement. This behaviour is compatible with our hypothesis that the extra variables are just functions of those first twenty variables (functions which are effectively re-implemented by the generative network).

Images generated by sampling from a Gaussian approximation of the latent distribution for different latent dimensions are shown in Fig. 4. Below dimension 20, most generated image can be recognized as a specific digit.

## 6 OUTLOOK

We studied the classical (non-quantum) form of the theory introduced in Bény & Osborne (2013), and found that the *relevant observables* of that theory are just the most correlated canonical variables in the sense of DCCA Andrew et al. (2013), and can be learned effectively using standard machine learning methods.

This point of views on DCCA provided us with several new insights. The first is that the learned relevant variables provide a useful representation of a joint probability distribution. We showed that performing inference using this representation can outperform crossentropy in predicting classes. Our experiments on halves of MNIST also show that the conditional distribution we obtain can effectively represent the uncertainty in the prediction of high-dimensional data.

A second insight relates to the interpretation of the canonical variables as spanning directions in the space of probability distributions. As suggested by the gaussian solutions and our experiment on synthetic data, we postulate that the canonical variables are functions of a small number of independent *generators* contained in their span. This hypothesis is supported by our experiment on MNIST, but further work is required to find a way to cleanly extract these variables.

We have yet to explore the potential applications of one of the salient aspect of this approach to inference, the fact that the canonical variables learned using DCCA are also those which can be most reliably predicted, irrespective of the value of the cutoff. To see why this is potentially significant, we observe that a central feature of scientific exploration is that we are not so concerned with making predictions about some given variables, as much as we are with discovering variables which can be predicted.

Another important feature of this approach is the fact that the resulting model allows for the direct evaluation of the expectation values in the posterior distribution without sampling. In particular this allows for the evaluation of credible intervals. Hence it should be especially suited to scientific applications where the ability to quantify uncertainty is essential.

Finally, the relationship that we established with theory of quantum origin points towards a potential quantum generalization of DCCA that would apply to quantum data, or classical measurements of quantum systems.

## ACKNOWLEDGMENTS

We would like to thank Joël Bény and Raban Iten for helpful suggestions. We are also indebted to an anonymous ICLR2020 referee for pointing out the connection between our approach and DCCA. This work was supported by the National Research Foundation of Korea (NRF-2018R1D1A1A02048436).

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

# A    EXTRA INFORMATION ABOUT THE ALGORITHM

## A.1    ALTERNATIVE INTERPRETATION OF THE OBJECTIVE

If we write $F_{ij} := f_j(x_i)$ and $G_{ij} := g_j(y_i)$ for the value of our variables on the dataset, then $K = \frac{1}{N}F^\top F, L = \frac{1}{N}G^\top G$ and $A = \frac{1}{N}G^\top F$. The DCCA objective can then be written as

$$\text{Tr}\left(K^{-1}A^\top L^{-1}A\right) = \text{Tr}\left(PQ\right)$$

where $P = F(F^\top F)^{-1}F^\top$ and $Q = G(G^\top G)^{-1}G^\top$ are the projectors on the ranges of $F$ and $G$ respectively. Hence, we are maximizing the overlap between those ranges (which represent possible linear combinations of datapoints, respectively determined from variables of one or the other correlated views.)

## A.2    HEURISTIC

**Batch size**—In our experiments, we observed that the batch size during training needs to be an order of magnitude larger than the number of variables (rank cutoff). When the batch size was too small, learning seemed to converge normally in terms of training and test loss, but resulted in variables which yield dramatically different losses when evaluated on larger batches, and yield spurious predictions.

**Constant variables**—The loss function $C$ takes value between $0$ and $k_0 - 1$ because the constant variable always has relevance $1$. The constant variable could be enforced a priori rather than learned, which, due to the objective, automatically forces the learned variables to have zero expectation values (be orthogonal to the constant variable). This might have advantages in certain circumstances, but in our experiments we found that this sometime hindered convergence.

**Invertibility issues**—The covariance matrices $K$ and $L$ can be ill-conditioned, potentially causing the gradient to "explode" because of the inverses $K^{-1}$ et $L^{-1}$ involved in the loss function. This can be avoided either by using the Moore-Penrose pseudo-inverse, or by replacing $K^{-1}$ by $(K + \epsilon\mathbf{1})^{-1}$ in the loss for some small positive number $\epsilon$, and likewise for $L^{-1}$.

**Symmetries in the loss function**—The loss $C$ only depends on the span of the variables $f_i$ and $g_j$, hence it has a very large group of symmetries. In particular, it is invariant under a change of the norm of each variable independently from each other. Because of that, it is preferable not to have a linear last layer. Using a hyperbolic tangent as last nonlinearity worked in our experiments.

**Regularization**—In all our tests, dropout had no beneficial effect. In fact, our objective seems to already provide a form of regularization, as shown in Section 5.2.

# B    THEORY IN MORE DETAILS

We consider two correlated random variables $X$ and $Y$ with a joint probability distribution $p(x, y)$. We assume that we are able to numerically evaluate expectations with respect to this distribution, for instance because we can sample from it. We want to use this ability in order to compute expectations with respect to the conditional distributions $p_{X|Y}(x|y) = p(x, y)/p_X(x)$ and $p_{Y|X}(y|x) = p(x, y)/p_Y(y)$, where $p_X(x) = \sum_y p(x, y)$ and $p_Y(y) = \sum_x p(x, y)$ are the marginals of $p$. Below we sometime remove the subscripts $X$, $X|Y$ or $Y|X$ if there is no ambiguity.

For instance, suppose we generated samples of $y$ given $x$, through explicit knowledge of $p_{Y|X}$. Then the evaluation of expectations with respect to $p_{X|Y}$ is the subject of Bayesian inference. However, this is generally done in a context where the variable $X$ has low dimensionality and parameterizes a hand-crafted model. Our approach, however, is free of such a model and the variable $X$ can be of very high dimensionality.

### B.1 INNER PRODUCT ON PROBABILITY VECTORS

In order to define our strategy, we need to equip the spaces of probability distributions for $X$ and $Y$ with an inner product structure. Let us focus on $X$, and assume that it takes discrete values to avoid unnecessary technicalities. The set of probability vectors is a convex subset of the real linear space $V_X = \mathbb{R}^n$. Let us equip this space with the product

$$\langle \mu, \mu' \rangle_X := \sum_x \frac{\mu(x)\mu'(x)}{p_X(x)} \tag{11}$$

for any $\mu, \mu' \in V_X$. We also write $\|\mu\|_X^2 = \langle \mu, \mu \rangle_X$. Importantly, this depends explicitly on the fixed probability vector $p_X(x)$, which we took to be the marginal of $p(x, y)$. If $p_X$ has full support, this makes $V_X$ into a real inner product space. The same can be done for the variable $Y$, yielding the inner product $\langle \nu, \nu' \rangle_Y$ for $\nu, \nu' \in V_Y$.

Had we interpreted $\mu$ and $\mu'$ as tangent vectors to $V_X$, considered as a manifold, this would be the Fisher information (Riemannian) metric, as in Bény & Osborne (2015b). But this quantity is also meaningful for finite vectors: the induced norm distance between $p_X$ and any probability vector $q$ is the $\chi^2$-divergence:

$$\chi^2(q, p_X) = \langle q - p_X, q - p_X \rangle_X. \tag{12}$$

The set of conditional probability distributions $p_{Y|X}$ form a stochastic map, i.e., a linear map $\mathcal{N} : V_X \to V_Y, \mu \mapsto \mathcal{N}(\mu)$, where

$$\mathcal{N}(\mu)(y) = \sum_x p_{Y|X}(y|x)\mu(x) \tag{13}$$

for any $\mu \in V_X$.

It is straightforward to check that the stochastic map $\mathcal{N}^*$ defined by

$$\mathcal{N}^*(\nu)(x) = \sum_x p_{X|Y}(x|y)\nu(x) \tag{14}$$

is the *transpose* $\mathcal{N}^*$ of $\mathcal{N}$ with respect to the inner products we defined (Ohya & Petz, 2004), i.e., for all $\nu \in V_Y$ and $\mu \in V_X$,

$$\langle \nu, \mathcal{N}(\mu) \rangle_Y = \langle \mathcal{N}^*(\nu), \mu \rangle_X. \tag{15}$$

Also, we observe that $\mathcal{N}(p_X) = p_Y$ and $\mathcal{N}^*(p_Y) = p_X$.

### B.2 EIGEN-RELEVANCE DECOMPOSITION

We can use the inner products on $V_X$ and $V_Y$ to define a singular value decomposition of the stochastic map $\mathcal{N}$. That is, there is an orthonormal family $u_1, \ldots, u_k$ of $V_X$ and an orthonormal family $v_1, \ldots, v_k$ of $V_Y$, such that

$$\mathcal{N}(u_j) = \eta_j v_j, \tag{16}$$

for $j = 1, \ldots, k$. For each $j$, $\eta_j$ is a singular value of $\mathcal{N}$, whose square we call the *relevance* of the vector $v_j$. Moreover $\eta_j \in [0, 1]$ since the $\chi^2$ divergence is contractive under any stochastic map. Given that $\mathcal{N}^*$ is the transpose of $\mathcal{N}$:

$$\mathcal{N}^*(v_j) = \eta_j u_j. \tag{17}$$

Equivalently, $u_j$ is an eigenvector of $\mathcal{N}^* \circ \mathcal{N}$ and $v_j$ is an eigenvectors of $\mathcal{N} \circ \mathcal{N}^*$, both with eigenvalue $\eta_j^2$.

Because $\mathcal{N}$ maps $p_X$ to $p_Y$, we always have the dual eigenvectors $u_0 = p_X$ and $v_0 = p_Y$ with eigenvalue 1.

### B.3 LOW-RANK APPROXIMATION

Typically, the dimension $k$ of the space of probabilities is more than astronomically large. For instance, if the values of $X$ consists of small 256 gray level images of $28 \times 28$ pixels, then $k =$

$256^{28^2} \simeq 10^{1888}$. However, in many case, only very few of these dimensions may be relevant for the purpose of inferring other variables.

The core of our approach is to approximate $\mathcal{N}$ and $\mathcal{N}^*$ by restricting them to the span of the first $k_0$ eigenvectors $u_j$ and $v_j$ with largest singular values $\eta_j$. That is, if we order the singular values $\eta_j$, $j = 1, \dots, k$ in decreasing order, we propose to use the approximations

$$\mathcal{N}_0(\mu) = \sum_{j \le k_0} \eta_j \langle u_j, \mu \rangle_X v_j \tag{18}$$

$$\mathcal{N}_0^*(\nu) = \sum_{j \le k_0} \eta_j \langle v_j, \nu \rangle_Y u_j \tag{19}$$

$$\tag{20}$$

to $\mathcal{N}$ and $\mathcal{N}^*$ respectively, for some $k_0$ typically much smaller than $k$, and any $\mu \in V_X, \nu \in V_Y$.

We denote the components of $\mathcal{N}_0$ and $\mathcal{N}_0^*$ by $q(y|x)$ and $q(x|y)$, e.g.,

$$\mathcal{N}_0(\mu)(y) = \sum_x q(y|x)\mu(x). \tag{21}$$

Since $\mathcal{N}_0$ and $\mathcal{N}_0^*$ are adjoint, we can define $q(x,y) = q(x|y)p_Y(y) = q(y|x)p_X(x)$. Although the marginals of $q(x,y)$ are the probability distributions $p_X$ and $p_Y$, the numbers $q(x,y)$ are not necessarily positive.

The quality of this approximation for a given $k_0$ does not directly depend on the dimensionality of $X$ and $Y$, but only on the amount of correlations between the two variables. Our aim is to use a $k_0$ small enough that the components of $\mathcal{N}_0$ and $\mathcal{N}_0^*$ can be computed explicitly.

**Theorem 1.** $\mathcal{N}_0$ *is the map of rank* $k_0$ *which minimizes the average distance*

$$\sum_x p(x)\|\mathcal{N}_0(\delta_x) - \mathcal{N}(\delta_x)\|_Y^2 = \sum_{xy} \frac{(q(x,y) - p(x,y))^2}{p(x)p(y)}. \tag{22}$$

*Proof.* The low rank approximation $\mathcal{N}_0$ minimizes the distance $\|\mathcal{N}_0 - \mathcal{N}\|_F$ where

$$\|\mathcal{M}\|_F^2 = \mathrm{Tr}\,(\mathcal{M}^*\mathcal{M}) \tag{23}$$

is the Hilbert-Schmidt (or Frobenius) norm (Eckart & Young, 1936). This follows from the fact that this is also the $l^2$-norm of the vector of singular values of $\mathcal{M}$. Let us find the explicit form of the trace. Each possible value $x$ of the variable $X$ is associated with a probability distribution $\delta_x(y) = 1$ when $x = y$ and zero otherwise. These distributions form an orthogonal basis of $V_X$, and have norms $\langle \delta_x, \delta_x \rangle = 1/p_X(x)$. Therefore,

$$\mathrm{Tr}\,(\mathcal{M}^*\mathcal{M}) = \sum_x p_X(x)\langle \delta_x, \mathcal{M}^*\mathcal{M}(\delta_x)\rangle_Y$$

$$= \sum_x p_X(x)\|\mathcal{M}(\delta_x)\|_Y^2$$

$\square$

## B.4 RELEVANT VARIABLES

We express the elements $\mu \in V_X$ and $\nu \in V_Y$ in terms of the marginals $p_X$ and $p_Y$ as simple products:

$$\mu(x) = p_X(x)f(x) \quad \text{and} \quad \nu(y) = p_Y(y)g(y) \tag{24}$$

for all $x, y$, where $f$ and $g$ are real functions of $x$ and $y$.

The inner products then simply become correlations among variables. Using also $\mu' = p_X f'$ and $\nu' = p_Y g'$, we obtain

$$\langle \mu, \mu' \rangle_X = \sum_x p_X(x)f(x)f'(x) = \overline{ff'}, \tag{25}$$

$$\langle \nu, \nu' \rangle_Y = \sum_y p_Y(y)g(y)g'(y) = \overline{gg'}. \tag{26}$$

These are simple expectation values with respect to $p$, which we assumed is the type of quantity we can evaluate for arbitrary functions $f, f', g, g'$.

Since $\mathcal{N}^*\mathcal{N}$ is self-adjoint in terms of this inner product, its eigenvectors $u_i$ are orthogonal, and hence the corresponding variables $a_i$ defined by $u_i(x) = p_X(x)a_i(x)$ are uncorrelated. Indeed,

$$\overline{a_i a_j} = \langle u_i, u_j \rangle_X = 0, \tag{27}$$

for all $i, j$. Moreover, accounting for the eigenvector $u_0 = p_X$ (corresponding to the constant feature $a_0(x) = 1$ for all $x$),

$$\overline{a_i} = 0 \tag{28}$$

for all $i \neq 0$. Hence we trivially have

$$\overline{a_i a_j} = \overline{a_i}\,\overline{a_j} \tag{29}$$

for all $i, j \neq 0$.

Likewise for the eigenvectors of $\mathcal{N}\mathcal{N}^*$. If $v_i(y) = p_Y(y)b_i(y)$:

$$\overline{b_i b_j} = \langle v_i, v_j \rangle_Y = 0 = \overline{b_i}\,\overline{b_j}. \tag{30}$$

for all $i, j \neq 0$.

Importantly, this does not mean that the variables $u_1, u_2, \ldots$ nor $v_1, v_2, \ldots$ are "disentangled", i.e., they are not statistically independent. These variables represent components in the space of probability vectors, rather than the "sample" space. They should be understood as spanning a subspace of the space of functions over the relevant independent variables. We discuss this in more detail in Section 5.3.

## B.5 Corners of $\mathcal{N}$ and loss function

The final piece of puzzle we need, is the ability to express the components (corners) of $\mathcal{N}$ and $\mathcal{N}^*$ in the span of possible non-orthogonal families of variables.

Let us therefore consider two arbitrary families $f_1, \ldots, f_{k_0}$ and $g_1, \ldots, g_{k_0}$ of variables, which respectively represent the vectors $p_X f_j \in V_X$ and $p_Y g_j \in V_Y$.

Firstly, we need matrices representing the components of the inner products on $V_X$ and $V_Y$. Those are the symmetric matrices

$$K_{ij} = \langle p_X f_i, p_X f_j \rangle_X = \overline{f_i f_j}, \tag{31}$$
$$L_{ij} = \langle p_Y g_i, p_Y g_j \rangle_Y = \overline{g_i g_j}. \tag{32}$$

The components $N_{ij}$ of $\mathcal{N}$ are defined by

$$\mathcal{N}(p_X f_j) = \sum_i N_{ij} p_Y g_i. \tag{33}$$

Taking the inner product with $p_Y g_k$, we obtain

$$\langle p_Y g_k, \mathcal{N}(p_X f_j) \rangle = \sum_i N_{ij} L_{ki}. \tag{34}$$

The left-hand side can be computed using Equ. 13. It is the matrix

$$\begin{aligned}
A_{kj} &= \langle p_Y g_k, \mathcal{N}(p_X f_j) \rangle \\
&= \sum_{x,y} \frac{p_Y(y)g_k(y)p_{Y|X}(y|x)p_X(x)f_j(x)}{p_Y(y)} \\
&= \sum_{x,y} p(x,y)g_k(y)f_j(x) = \overline{g_k f_j}.
\end{aligned} \tag{35}$$

Therefore, in matrix notation, Equ. (34) is $A = LN$, or

$$N = L^{-1}A. \tag{36}$$

The components $N_{ij}^*$ of $\mathcal{N}^*$ are obtained by just swapping $X$ and $Y$, yielding

$$N^* = K^{-1}A^\top. \tag{37}$$

Hence the singular values of the corner of $\mathcal{N}$ defined by the variables $f_j$ and $g_j$ are just the square-root of the eigenvalues of the matrix $N^*N = K^{-1}A^\top L^{-1}A$. In order to find the variables $f_j$ and $g_j$ with the same span as the first $k_0$ eigenvectors $u_j$, $v_j$, we just need to maximize all the eigenvalues of $N^*N$. A simple way to do this is to use (minus) the trace of $N^*N$ as loss function, since it is the sum of the square of the singular values. We call $\mathrm{Tr}\,(N^*N)$ the *relevance* of the subspaces defines by the variables $f_j$ ad $g_i$ for all $i, j$. This yields the loss/cost function:

$$C = k_0 - \mathrm{Tr}\,(N^*N) = k_0 - \mathrm{Tr}\,(K^{-1}A^\top L^{-1}A). \tag{38}$$

Once optimal variables have been found, one can obtain the components of the eigenvectors in the span of $f_1, \ldots, f_{k_0}$ through standard numerical diagonalization of $N^*N$.

## B.6 INFERENCE

The variables minimizing $C$ can be used to infer one variable from the other. For instance, given $y$, the inferred probability distribution over $x$ is given by $p_{X|Y}(x|y) = \mathcal{N}^*(\delta_y)(x)$, where $\delta_y(y')$ is 1 when $y = y'$ and zero otherwise. In order to compute this, we first need the components of the distribution $\delta_y$ in terms of the family $p_Y g_1, \ldots, p_Y g_{k_0}$, i.e., the real numbers $(\delta_y)_j$ such that

$$\delta_y(y') = p_Y(y') \sum_{i=1}^{k_0} (\delta_y)_i g_i(y') + r(y'), \tag{39}$$

where $\langle r, p_Y \delta_i \rangle_Y = 0$ for all $i$. Taking the inner product with $p_Y g_j$, we obtain

$$\langle p_Y g_j, \delta_y \rangle_Y = \sum_{i=1}^{k_0} (\delta_y)_i L_{ji}, \tag{40}$$

where the left hand side is also just

$$\langle p_Y g_j, \delta_y \rangle_Y = g_j(y). \tag{41}$$

Therefore the components of $\delta_y$ are explicitly

$$(\delta_y)_i = \sum_j (L^{-1})_{ij} g_j(y). \tag{42}$$

It follows that

$$\begin{aligned} p_{X|Y}(x|y) = \mathcal{N}^*(\delta_y)(x) &\approx \mathcal{N}_0^*(\delta_y)(x) \\ &= \sum_{ijk} N_{ki}^* (L^{-1})_{ij} g_j(y) f_k(x). \end{aligned} \tag{43}$$

Then, for instance, the expected inferred value of $X$ is

$$\overline{x} = \sum_{ijk} N_{ki}^* (L^{-1})_{ij} g_j(y) \sum_x p_X(x) x f_k(x). \tag{44}$$

For the inference of $Y$ from $x$, we have

$$p_{Y|X}(y|x) \approx \sum_{ijk} N_{ki} (K^{-1})_{ij} f_j(x) g_k(y). \tag{45}$$

## C   ANALYTICAL EXAMPLE

When $p(x, y)$ is any multivariate Gaussian distribution, everything can be computed analytically. Let us consider here the one-dimensional case. We use $p(x) \propto \exp\left(-x^2/2\tau^2\right)$, and the conditional $p(y|x) \propto \exp\left(-(y-x)^2/2\sigma^2\right)$. That is, $y$ is equal to $x$ but with some added Gaussian noise. This gives

$$p_{X|Y}(x|y) \propto \exp\left(-\frac{(x-\gamma y)^2}{2\tau^2(1-\gamma)}\right), \quad \text{where} \quad \gamma = \frac{\tau^2}{\sigma^2+\tau^2}. \tag{46}$$

It was show in Lancaster (1958), that the most relevant subspace of dimension $k_0$ on the variable $X$ is simply spanned by the variables

$$f_n(x) = x^n, \tag{47}$$

$n = 0, \ldots, k_0 - 1$. Similarly for $Y$;

$$g_n(y) = y^n. \tag{48}$$

This independence of the relevant variables on the detailed parameters of $p$ is a general property of Gaussian joint distributions.

This means, for instance, that the most relevant feature ($n = 1$) for predicting the value of $X$ given $Y = y$ is simply $Y$ itself. The higher order variables have to do with inferring extra aspects of the probability distribution over $X$.

A set of orthogonal variables can be obtain from the Gram-Schmidt procedure, which, if done from small to large $n$ much necessarily yield the eigenvectors $u_n$ and $v_n$. For illustration purpose, let us work with the non-orthogonal vectors $f_n$ and $g_n$, keeping only the first $k_0 = 3$ vectors.

The three matrices (correlators) we need can be easily computed:

$$K = \begin{pmatrix} 1 & 0 & \tau^2 \\ 0 & \tau^2 & 0 \\ \tau^2 & 0 & 3\tau^4 \end{pmatrix} \qquad L = \begin{pmatrix} 1 & 0 & \tau^2+\sigma^2 \\ 0 & \tau^2+\sigma^2 & 0 \\ \tau^2+\sigma^2 & 0 & 3(\tau^2+\sigma^2)^2 \end{pmatrix} \tag{49}$$

$$A = \begin{pmatrix} 1 & 0 & \tau^2 \\ 0 & \tau^2 & 0 \\ \tau^2+\sigma^2 & 0 & \tau^2(\sigma^2+3\tau^2) \end{pmatrix}. \tag{50}$$

We obtain

$$M = K^{-1}A^\top L^{-1}A = \begin{pmatrix} 1 & 0 & \tau^2(1-\gamma^2) \\ 0 & \gamma & 0 \\ 0 & 0 & \gamma^2 \end{pmatrix}. \tag{51}$$

The eigenvalues of $M$ can be read on the diagonal, and the corresponding eigenvectors are $(1, 0, 0)$, $(0, 1, 0)$ and $(-\tau^2, 0, 1)$, which means that the eigenfunctions are in order $u_0(x) = 1$, $u_1(x) = x$ and $u_2(x) = x^2 - \tau^2$.

Because we are working with continuous variables, the true rank of $\mathcal{N}$ is infinite, even for any finite cutoff on the singular values. Nevertheless, it is instructive to see how the approximate inference fares for rank $k_0 = 3$. Given the value $y$ for $Y$, the inferred distribution over $X$ is

$$\mathcal{N}_0^*(\delta_y)(x) = p_X(x)p_Y(y) \sum_{j,k=0}^{2} (K^{-1}A^\top L^{-1})_{kj} y^j x^k. \tag{52}$$

The approximately inferred first and second moments of $X$ is given by integrating the above times $x$ (resp. $x^2$) over $x$. We obtain

$$\overline{x} = \gamma y \quad \text{and} \quad \overline{x^2} = \gamma^2 y^2 + (1-\gamma)\tau^2, \tag{53}$$

which are actually exact: they are equal to the first two moments of $X$ over $p_{X|Y}$ as given in Eq. (46).

In fact, it is easy to see that this would be true for the first $k_0 - 1$ moments had we kept the $k_0$ most relevant variables.

