# OpenReview forum: "Learning relevant features for statistical inference"
_ICLR.cc/2020/Conference — Reject_

### Official Review · AnonReviewer3 · 2019-10-20
**Official Blind Review #3**

**Rating:** 3

**Review:**

The paper proposes how the correlation between two different types of data can be extracted from learned representations. The proposed metric can also be used as an alternative to cross entropy loss. The paper provides analytical calculations as well as real data sets simulations/experiments. However there are significant draw-becks:

1) Similarity and Metric Learning is a booming area in machine learning with several different directions focusing on different problems. The paper fails to locates itself in the literature, how it compares itself into other techniques (both analytically and experimentally).

2) The proposed technique seems to be very similar to SVD of learned representations. Connection to quantum field theory is well established but more simpler comparisons to SVD and other spectral techniques are not provided in metric learning.

3) Novelty is not clear. There are interesting experiments in disentangled feature feature extraction and data generation. However, they are mostly proof of concept and lack of baselines. It is not clear what problem this technique solves better compared to other existing solutions.

Paper is mostly written clearly. I do suggest putting Appendix A1 to the main paper though.

**Experience Assessment:**

I have published one or two papers in this area.

**Review Assessment: Checking Correctness Of Derivations And Theory:**

I assessed the sensibility of the derivations and theory.

**Review Assessment: Checking Correctness Of Experiments:**

I assessed the sensibility of the experiments.

**Review Assessment: Thoroughness In Paper Reading:**

I read the paper at least twice and used my best judgement in assessing the paper.

---

### Official Review · AnonReviewer1 · 2019-10-23
**Official Blind Review #1**

**Rating:** 3

**Review:**

The paper considers how to learn correlations between two spaces, e.g., input/output, in order to generate data in one space conditioned on values from the other. This is performed by modeling features with neural networks and optimizing an objective function that maximizes a measure of correlation between the features versus learning a generative model such as a CVAE. Some illustrative examples using MNIST are provided.

My decision is to reject. I think there is value in the approach, but it is hard to see clearly at the moment given that the exposition is difficult to follow and the experiments aren't very compelling. If these issues could be addressed (concrete suggestions below), and some of the follow-on work in the last section could be performed, I think there could be a pretty interesting contribution here.

***

Decision-related suggestions/questions:

* Include more datasets in the experimental section. The second sentence of the introduction lists possibilities such as time series and multi-modalities that I would have been very interesting.

* The first claim that there is no tunable variable in the objective function is a little hard to parse. Clearly, the rank of the low-rank approximation must be set, and the features of the two spaces need to be learned. Some clarification here would be helpful.

* There are a number of unfortunate typos/grammar issues/presentation choices that really impact clarity. Some examples:
	* In the third paragraph under theory, "...linear spaces spanned by the probability distributions..." should probably be "...linear spaces spanned by all probability distributions..." (?)
	* The following sentence is a run-on.
	* In (8), occurrences of g_*(x_n) should be replaced with g_*(y_n).
	* The replacement of the (low) rank symbol, k_0, with the sample size symbol, n, in the second paragraph of section 4.1.
	* Introducing a "Bayesian estimator for an l^2 distance" w/o explanation. What does this mean?

* How should the low-rank parameter k_0 be selected generally given that the singular value distribution may not always be useful in selecting it?

* Can anything be said quantitatively or qualitatively about the sample complexity required to estimate the matrices of (8) well enough to estimate the features?

* Is there an interpretation for why both spaces require the same feature dimension, k_0?

***

Comments not related to decision:

* It is generally good to avoid sweeping statements such as the first sentence of the introduction. Perhaps consider replacing with a simple statement on the intended goal of the paper: "...produce a useful model of correlations... for the task of data generation..."

* Consider placing a concrete, motivating example prior to the theory section as it is hard to digest (from an ML perspective) without a clear context. The analytical example with the Gaussian from the supplementary material is one option.

* The last statement of the paragraph under (3) needs a reference.

* It seems strange to have the supervised learning experiment of 4.1 as the first experimental result of the paper since it is an unintended and unexplained side-effect of the approach. Also, the claim of "faster convergence" should be demonstrated in wall-clock time.

**Experience Assessment:**

I do not know much about this area.

**Review Assessment: Checking Correctness Of Derivations And Theory:**

I assessed the sensibility of the derivations and theory.

**Review Assessment: Checking Correctness Of Experiments:**

I assessed the sensibility of the experiments.

**Review Assessment: Thoroughness In Paper Reading:**

I read the paper at least twice and used my best judgement in assessing the paper.

---

> ### Author Response · Authors · 2019-11-13
> **Answers**
>
> > * Include more datasets in the experimental section. The second sentence of the introduction lists possibilities such as time series and multi-modalities that I would have been very interesting.
>
> I agree, but that's still work in progress. I thought that the theoretical contribution was sufficiently interesting to warrant this submission.
>
>
> > * The first claim that there is no tunable variable in the objective function is a little hard to parse. Clearly, the rank of the low-rank approximation must be set, and the features of the two spaces need to be learned. Some clarification here would be helpful.
>
> The objectives for VAE and related variational algorithms are linear combinations of two or more competing losses. I was referring to the "Lagrange parameters" in these linear combinations. They complicate the learning process, as they must be set to carefully balances to different losses, and there have no known correct theoretical values.
>
> The rank cutoff has a known theoretical best value: the larger the better. Also it is rather easy to see from the obtained spectrum whether it is too small or unnecessarily large (by checking if the spectrum reaches zero).
>
>
> > * There are a number of unfortunate typos/grammar issues/presentation choices that really impact clarity...
>
> Thanks, I fixed those in the latest draft.
>
>
> > * How should the low-rank parameter k_0 be selected generally given that the singular value distribution may not always be useful in selecting it?
>
> In the latest revision, I added a graphs of the singular values obtained after 150 epochs for different values of the cutoff k_0. I think it gives a good idea of the typical behaviour I've observed so far. We can see that the spectrum converges, with typically less reliability on the smallest eigenvalues.
>
>
> > * Can anything be said quantitatively or qualitatively about the sample complexity required to estimate the matrices of (8) well enough to estimate the features?
>
> That's an important question, but I haven't done a theoretical analysis of this aspect so far. In doi:10.1093/imaiai/ias001  the authors state that the number of sample needed to estimate a covariance matrix (as measured in spectral norm and assuming a Gaussian distributions) goes as k_0/eps^2, where k_0 is the dimension and eps a bound on the spectral distance.  But I'm not sure what value of eps is adequate for DCCA.  In my experiments, it appeared to work as long as the batch size were at least 10-20 times larger than the dimension k_0.
>
> > * Is there an interpretation for why both spaces require the same feature dimension, k_0?
>
> Yes: the rank of the effective representation of the channel N is bounded by the smallest of the dimensions, so the difference between the two ranks would just yield as many zero singular values.
>
>
> > * Consider placing a concrete, motivating example prior to the theory section as it is hard to digest
>
> The new version of the introduction should be more clear.
>
>
> > * The last statement of the paragraph under (3) needs a reference.
>
> See for instance https://en.wikipedia.org/wiki/Singular_value_decomposition.
>
>
> > * It seems strange to have the supervised learning experiment of 4.1 as the first experimental result of the paper since it is an unintended and unexplained side-effect of the approach. Also, the claim of "faster convergence" should be demonstrated in wall-clock time.
>
> I changed the order in the new draft.
>
> I added a comment to the effect that epoch is a fair proxy for clock time in that case.

---

### Official Review · AnonReviewer2 · 2019-10-26
**Official Blind Review #2**

**Rating:** 1

**Review:**

The paper proposes an approach to find a map between two feature spaces to maximize correlation between them and to use the resulting map for inference. A theoretical exposition is given and some empirical results are provided showing that the approach speeds up convergence on supervised MNIST and can be used for image completion (again on MNIST).

The paper should be rejected for the following reasons. First, the approach looks very similar to deep CCA, but the connection is never mentioned. This connection needs to be clarified. The objective function needs to be clearly stated and related to the loss function in eq. (7). In particular, I would suggest to give a clear definition of the problem before delving into the theory in section 2. In its current version, it is difficult to assess how the parts of section 2 relate to the overall objective. The paper severely lacks in relation to relevant related work. Half(!) of the 14 referenced papers are by the author himself. This can be verified since the double blind review process is compromised as the paper links to code in the author’s public github account. Finally, the empirical results are quite incomplete. It is not clear how the results compare to generative methods like VAEs, which are referenced as a motivation for this work in this work.

The improvement in convergence from RFA for supervised learning is interesting and this aspect deserves more analysis. It would be useful to look at the total amount of computation required to reach a given loss. I also wonder how this differs from simply mapping the output of the first network to a low-rank space via PCA. Is the dual-view really necessary in this case since the information content in the label space must be very limited, beyond simple class balance statistics?


**Experience Assessment:**

I have read many papers in this area.

**Review Assessment: Checking Correctness Of Derivations And Theory:**

I assessed the sensibility of the derivations and theory.

**Review Assessment: Checking Correctness Of Experiments:**

I assessed the sensibility of the experiments.

**Review Assessment: Thoroughness In Paper Reading:**

I read the paper at least twice and used my best judgement in assessing the paper.

---

> ### Author Response · Authors · 2019-11-13
> **Answers from author**
>
> Indeed, my objective function turns out to be the same as that of deep CCA. This is addressed in the revised submission.
>
> > I would suggest to give a clear definition of the problem before delving into the theory in section 2. In its current version, it is difficult to assess how the parts of section 2 relate to the overall objective.
>
> I hope the my new introduction and theory section clarifies these aspects.
>
>
> > The objective function needs to be clearly stated and related to the loss function in eq. (7).
>
> The objective function is precisely Eq. (7). I changed the terminology; assuming this was the source of your doubts.
>
>
> > ... the double blind review process is compromised as the paper links to code in the author’s public github account.
>
> Sorry for that. My reasoning was that the anonymity was compromised anyway by the paper being on the arxiv, but I see now that would have still required an active search at least.
>
>
> > Finally, the empirical results are quite incomplete. It is not clear how the results compare to generative methods like VAEs, which are referenced as a motivation for this work in this work.
>
> I agree that more comparisons would be useful.
>
> The reason I didn't do any heads-on comparison about the quality of the latent variables so far stems from the actual differences between the two approaches. My approach doesn't work as an autoencoder for instance, which prevents the more objective benchmarks (unsupervised pretraining). Moreover, it also doesn't naturally produces samples, which also prevents standard qualitative tests. Although I tried to go around this by using noisy images in my last experiments, I thought the result wasn't good enough to pursue this route. But I am exploring different ideas at the moment.
>
> Also I'm not sure yet how to directly compare the ability of this approach to model the full conditional distribution against other methods.
>
>
> > The improvement in convergence from RFA for supervised learning is interesting and this aspect deserves more analysis. It would be useful to look at the total amount of computation required to reach a given loss.
>
> Yes I should have specified that the amount of computation, or clock time, is essentially the same for all approaches. Indeed, only the objective is different, and the computational cost is dominated by the forward and backward passes through the neural net. This may not be the case for larger number of classes, however, mainly because this may require larger batches. I added a paragraph explaining this.
>
>
> > I also wonder how this differs from simply mapping the output of the first network to a low-rank space via PCA. Is the dual-view really necessary in this case since the information content in the label space must be very limited, beyond simple class balance statistics?
>
> I tried to explain this in the first paragraph of that section. The dual view is not necessary in the sense that we need only one neural net, with the exact same architecture as for standard supervised learning (with one tanh output per category). The second network can be replaced by the one-hot encoding function on the labels, which is optimal. The DCCA objective doesn't simplify very much if this is hand-coded into it, and I'm not sure how it would relate to PCA. Also, a second (fast) inference step is still needed.

---

### Author Response · Authors · 2019-11-11
**Revision #1**

As pointed out by Reviewer #2, I had regrettably originally missed the existence of DCCA, whose objective turns out to be precisely the same as the one I derived in this submission. I would like to apologize to the authors of the DCCA and related papers, in case they are seeing this.

This new draft is thoroughly rewritten to try accounting for the related literature, or what I had time to review so far at least. I may still implement more changes before the deadline, including directly addressing the reviewer comments. But I want to submit this for now so that the reviewers may have some time to potentially give more feedback.

I completely rewrote the abstract, introduction, theory and conclusion, and added a section on related work.

I replaced the results on the occluded MNIST digits with ones which makes the multiple modes more apparent. The improvement was obtained by augmenting the data with slight rotations and translations, which increases the ambiguities in the predictions.

---

> ### Comment · AnonReviewer2 · 2019-11-11
> **Improved but still feels like work in progress**
>
> The new draft is certainly an improvement over the previous version. However, this still feels like work in progress. I would encourage the authors to flesh out this work more fully and map out all the connections to relevant work. I would also encourage more focused experiments with more detail. There seems to be many interesting aspects of this approach and a conference paper only has a limited number of pages...

---

### Decision · Program_Chairs · 2019-12-19

**Decision:**

Reject

**Comment:**

This manuscript proposes an approach for estimating cross-correlations between model outputs, related to deep CCA. Authors note that the procedure improves results when applied to supervised learning problems.

The reviewers have pointed out the close connection to previous work on deep CCA, and the author(s) have agreed. The reviewers agree that the paper has promise if properly expanded both theoretically and empirically.